# Radiotherapy in Medulloblastoma—Evolution of Treatment, Current Concepts and Future Perspectives

**DOI:** 10.3390/cancers13235945

**Published:** 2021-11-26

**Authors:** Clemens Seidel, Sina Heider, Peter Hau, Annegret Glasow, Stefan Dietzsch, Rolf-Dieter Kortmann

**Affiliations:** 1Department of Radiation Oncology, University Hospital Leipzig, 04103 Leipzig, Germany; sina.heider@medizin.uni-leipzig.de (S.H.); Annegret.Glasow@medizin.uni-leipzig.de (A.G.); stefan.dietzsch@medizin.uni-leipzig.de (S.D.); Rolf-Dieter.Kortmann@medizin.uni-leipzig.de (R.-D.K.); 2Wilhelm Sander-NeuroOncology Unit, Regensburg University Hospital, 93053 Regensburg, Germany; Peter.Hau@klinik.uni-regensburg.de

**Keywords:** medulloblastoma, radiotherapy, craniospinal irradiation, neurotoxicity

## Abstract

**Simple Summary:**

Craniospinal irradiation (CSI) is the backbone of medulloblastoma treatment and the first treatment to achieve a cure in many patients. Within the last decades, significant efforts have been made to enhance efficacy in combination with chemotherapy. With this approach, a majority of low- and standard-risk patients can be cured. In parallel, many clinical trials have dealt with CSI-dose reduction and reduction of boost volume in order to decrease long-term toxicity, particularly neurotoxicity. Within these trials, standardized quality assurance has helped to increase the accuracy of treatment and improve prognosis. More recently, advances of radiotherapy techniques such as proton treatment allowed for better sparing of healthy tissue in order to further diminish detrimental long-term effects. Major future challenges are the adaption of radiotherapy regimens to different molecularly defined disease groups alone or together with new targeted agents. Moreover, and even more importantly, innovative combinatorial treatments are needed in high- and very-high risk situations.

**Abstract:**

Medulloblastoma is the most frequent malignant brain tumor in children. During the last decades, the therapeutic landscape has changed significantly with craniospinal irradiation as the backbone of treatment. Survival times have increased and treatments were stratified according to clinical and later molecular risk factors. In this review, current evidence regarding the efficacy and toxicity of radiotherapy in medulloblastoma is summarized and discussed mainly based on data of controlled trials. Current concepts and future perspectives based on current risk classification are outlined. With the introduction of CSI, medulloblastoma has become a curable disease. Due to combination with chemotherapy, survival rates have increased significantly, allowing for a reduction in radiation dose and a decrease of toxicity in low- and standard-risk patients. Furthermore, modern radiotherapy techniques are able to avoid side effects in a fragile patient population. However, high-risk patients remain with relevant mortality and many patients still suffer from treatment related toxicity. Treatment needs to be continually refined with regard to more efficacious combinatorial treatment in the future.

## 1. Introduction

Medulloblastoma is the most frequent malignant brain tumor in childhood while it is very rare in adults [1]. Approximately 0.5–1 in 100,000 children are affected by the disease [1]. Within the past decades, major developments in combined radiochemotherapy have dramatically altered the therapeutic landscape of medulloblastoma. Prognosis increased in low- and moderate-risk patients to >80% long term survival [2,3,4]. However, a significant proportion of patients, particularly with high-risk features, remains uncured [3] or bears relevant toxicity after curative treatment.

Within this review, current clinical evidence and evolution of treatment concepts concerning the use of radiotherapy in medulloblastoma are outlined with a focus on the data of larger controlled clinical trials. Furthermore, general aspects of risk stratification and future concepts based on new molecularly based risk stratification are also outlined.

## 2. Methods

A PubMed search was performed by applying the keywords “medulloblastoma and radiotherapy” and “medulloblastoma and neurotoxicity” with a focus on primary or secondary results of controlled clinical trials and large series. Concerning “early” treatments before 1990 and techniques of special interest such as proton treatment and also smaller series have been included. In addition, recent review articles and relevant literature have been compiled regarding risk stratification and new combinatorial treatments in the treatment of medulloblastoma.

## 3. Results of Clinical Trials

Results of the most randomized controlled trials regarding the use of radiochemotherapy in children with medulloblastoma and use of proton treatment are described in chronological order and summarized in Table 1 and Table 2.

### 3.1. Radiation as Monotherapy—Lessons from Early Times

In the 1960s and 1970s, several retrospective series described varying 5-year survival rate of 12–50% after surgery and radiotherapy for medulloblastoma [5,6,7,8]. Survival was mainly short due to high post-operative mortality and many children did not accomplish radiotherapy. Bloom et al. reported a higher survival rate of 38% of patients completing radiotherapy compared to 32% in children with incomplete radiotherapy [5].

The most promising series from Jenkin et al. described a retrospective series of 47 children from Toronto covering two time periods from 1940–1952 and from 1953–1965. Within the latter period, it was noticed that treatment with CSI instead of local radiotherapy was able to prevent spinal relapse and that a completed CSI was the first curative treatment with 8/15 patients living more than five years after completion of radiotherapy [8].

In a later larger update from the Toronto series, 122 patients with medulloblastoma received radiotherapy after tumor resection between 1958–1978. Overall 5- and 10-year survival rates were 56% and 43%, respectively. A total of 46% of patients relapsed first in the posterior fossa. Fifteen patients who received a posterior fossa dose of 52 Gy or more after a total resection had a 5-year and 10-year survival rate of >75 [9].

Already in the early 1990s, reduction in treatment dose appeared desirable. However, a combined Children’s Cancer Group-Pediatric Oncology Group study including 126 patients with low-stage medulloblastoma comparing two different doses of neuroaxis irradiation (36 Gy in 20 fractions vs. 23.4 Gy in 13 fractions) led to early study termination after 16 months as a statistically significant increase was observed in the number of all relapses as well as isolated neuroaxis relapses in patients randomized to the lower dose of neuroaxis radiation [9,10].

**Table 1 cancers-13-05945-t001:** Efficacy of radiochemotherapy in medulloblastoma in children—results of large randomized trials.

Author (Year)	Trial	Patient Number	Inclusion Criteria (Age/KPS)	Chemotherapy	Radiotherapy	Endpoint (PFS/OS)	Remarks/Neurotoxicity
Tait, D.M. 1990 SIOP I [11]	Prospective, randomized	*N* = 268	Age less than 16 years, histopathological cerebellar medulloblastoma, or high-grade intracranial ependymoma	During RT VCR, maintenance CCNU/VCR for 1 year in 6 week cycles, 164 patients with chemotherapy	CSI ED: 35–45 Gy boost PF ED: 50–55 Gy	5y-OS 53% 5y-DFS 48% 10y-OS 45%	Significant difference in DFS in favor of chemotherapy, the difference declines over the years
Evans, A.E. 1990 [12]	Prospective, randomized	*N* = 233	Between 2 and 16 years of age, histologically proven medulloblastoma	Concurrent CCNU, followed by 1 year CCNU/VCR/Prednisone in cycles lasting 6 weeks	CSI ED: 35–45 Gy boost PF ED: 50–55 Gy	RCH: 5y-EFS 59% 5y-OS 65% RT: 5y-EFS 50% 5y-OS 65%	In patients with more extensive tumors, EFS was better in the group receiving chemotherapy (48% vs. 0%, *p* = 0.006)
Packer, R.J. 1994 [13]	Prospective study	*N* = 63	Older than 18 months of age, high-risk	VCR weekly during radiotherapy followed by 8 × 6-week cycles of Cisplatin/CCNU/VCR	Standard CSI ED: 36.0 Gy boost TB ED: 54.0–55.8 Gy RD: CSI ED: 23.4 Gy boost TB ED: 54.0 Gy, SD 1.8 Gy	5y-PFS: 85.0% 5y-OS: 66.0% M1-3: 5y-PFS: 67.0% M0: 5y-PFS: 90.0%	47.6% significant ototoxicity grade 3 to 4 renal tox. in 13 pts. grade 3 or 4 hematotox. 33/63 pts.
Bailey, C.C. 1995 SIOP II [14]	Prospective randomized	*N* = 364	Children with total or subtotal removal of the tumor	Low-risk: VCR/MTX/Procarbazine 6-weeks before RT high-risk: VCR/CCNU after RT	Low-risk: standard CSI ED: 35 Gy, SD: 1.66 Gy reduced CSI ED: 25 Gy + boost PF ED: 55 Gy	5y EFS 58.9% high-risk: 5y EFS 56.3%	No benefit for chemotherapy for any group, poor outcome in patients after chemotherapy and reduced dose radiotherapy
Packer, R.J. 1999 [15]	Prospective, non-randomized study	*N* = 65	Age 3–10 years with nondisseminated MB	CCNU/VCR/Cisplatin during and after RT	CSI ED: 23.4 GY, SD: 1.8 Gy boost TB ED: 55.8 Gy	3y-PFS: 86.0% 5y-PFS: 79.0%	Cisplatin dose had to be modified in more than 50% of pat. before the completion of treatment
Kortmann, R.D. 2000 HIT 91 [16]	Prospective, randomized	*N* = 137	Children between 3 and 18 years of age	Arm 1 (*N* = 72) neoadjuvant IFO/ETO/HD-MTX/ Cisplatin/Cytarabine arm 2 (*N* = 65) conc. VCR + Cisplatin/CCNU/VCR	CSI ED: 35.2 Gy, SD: 1.6 Gy boost PF ED: 55.2 Gy, SD: 2 Gy	all: 3y-PFS 66% R0: 3y-PFS 72% M2/3: 3y-PFS 30%	Negative prognostic factors were M2/3 disease, maintenance chemotherapy appears more effective in low-risk medulloblastoma
Taylor, R.E. 2004 SIOP PNET3 [17]	Prospective randomized study	*N* = 179	Age between 3 and 16 years, histologically proven MB, absence of leptomeningeal metastases on spinal MRI	A: RT alone B: RT + VCR/ETO/Carboplatin/Cyclo	CSI ED: 35 Gy, SD: 1.7 Gy, boost PF ED: 55 Gy	5y-OS: 70.7% 5y-EFS: 67.0% A: 3y-EFS: 64.8% B: 3y-EFS: 78.5% (*p* = 0.0366)	Multivariate analysis identified the use of chemotherapy (*p =* 0.0248) and RT duration (*p =* 0.0100) as predictive of better EFS
Packer, R.J. 2006 [2]	Prospective randomized study	*N* = 421	histologically confirmed MB, age 3–21 at time of diagnosis	I: CCNU/Cisplatin/VCR II: Cisplatin/VCR/Cyclo	CSI ED: 23.4 Gy, SD: 1.8 Gy, boost PF ED: 55.8 Gy	5y-OS: 81.0%, 86.0%	infections occurred more frequently in the Cyclophosphamide arm
Hoff, K. 2009 HIT’91 [18]	Prospective randomized study	*N* = 280	Patients with medulloblastoma (3–18 years) included from 1991 to 1997 in the randomized multicenter trial HIT’91	VCR concomitant with RT maintenance CCNU/VCR/Cisplatin Sandwich: two courses, each four cycles of IFO/ETO/HD-MTX/Cisplatin/Cytarabine	CSI ED: 35.2 Gy SD: 1.6 Gy boost PF ED: 55.2 Gy	Maintenance: M0 10y-OS 91% M1 10y-OS 70% M2/3 10y-OS 42% sandwich treatment: M0 10y-OS 62% M1 10y-OS 34% M2/3 10y-OS 45%	Long-term analysis, incomplete staging, metastases, younger age and sandwich chemotherapy were independent adverse risk factors
Lannering, B. 2012 SIOP PNET 4 [19]	Randomized multicenter trial	*N* = 340	Age 4 to 21	During RT VCR weekly adjuvant chemotherapy 6 weeks after RT, 8 cycles Cisplatin/CCNU/VCR with 6-week interval between each cycle	standard CSI ED: 23.4 Gy, boost PF ED: 55.8 Gy SD: 1.8 Gy hyperfractionated CSI ED: 36.0 Gy, SD: 1.0 Gy 2x/day boost PF ED: 60.0 Gy boost TB ED: 68.0 Gy	Standard: 5y-EFS 77.0% 5y-OS 87.0% hyperfractionated: 5y-EFS 78.0% 5y-OS 85.0%	Residual tumor of more than 1.5 cm^2^ was negative prognostic factor, severe hearing loss was not different between arms

Abbreviations: craniospinal irradiation (CSI), radiotherapy (RT), posterior Fossa (PF), tumorbed (TB), single dose (SD), end dose (ED), reduced dose (RD), Ifosfamide (IFO), Etoposide (ETO), Vincristine (VCR), Lomustine (CCNU).

**Table 2 cancers-13-05945-t002:** Clinical trials and series applying proton radiotherapy in medulloblastoma.

Author (Year)	Trial	Patient Number	Inclusion Criteria (Age)	Chemotherapy	Radiotherapy	Endpoint (PFS/OS)	Remarks/Neurotoxicity
Eaton, B.R. 2016 [20]	Multi-institution cohort study prospectively in enrolled in the Phase II study	*N* = 88 *x n*= 43 *p n* = 45	SR patients, age >3 years, <1.5 cm^2^ residual disease, M0	Adjuvant VCR/Cisplatin/Cyclo and/or CCNU	x-CSI ED: 23.4 Gy boost PF or TF ED: 54–55.8 Gy SD: 1.8 Gy p-CSI ED: 23.4 Gy boost PF or TF ED: 54–55.8 Gy	p6y-OS: 82.0% x6y-OS: 87.6% p6y-PFD: 78.8% x6y-PFS: 76.5%	Disease control with proton and photon radiotherapy appears equivalent for SR MB
Yock, T.I. 2016 [21]	Non-randomized, open-label, single-center, phase 2 single-arm study	*N* = 59	Age 3–21 years with MB	Chemotherapy before, during, or after RT with dose reductions for toxic effects. Total cumulative Cisplatin dose was recorded	passively scattered p-RT CSI ED: 18–36 Gy SD: 1.8 Gy	3y-PFS: 83.0%	45 evaluable patients had grade 3–4 ototoxicity cumulative incidence of any neuroendocrine deficit at 5 years was 55% (95% CI 41–67), with growth hormone deficit being most common
Brown, A.P. 2013 [22]	Retrospective review	*N* = 40	Adults with histologically confirmed MB, treated consecutively with CSI 16 years or older	All patients concurrent chemotherapy	x-CSI (*N* = 21) ED: 30.6 Gy boost TB or PF ED: 54 Gy p-CSI (*N* = 19) ED: 30.6 Gy boost TB or PF ED: 54 Gy	not evaluated	Patients treated with p-CSI experienced less treatment-related morbidity including less acute gastrointestinal and hematologic toxicities

Abbreviations: craniospinal irradiation (CSI), radiotherapy (RT), posterior Fossa (PF), tumorbed (TB), single dose (SD), end dose (ED), Vincristine (VCR), x-(Photon), p-(Proton).

### 3.2. Combining Radiotherapy and Chemotherapy—The Major Step toward Cure

After initial promising reports regarding efficacy of chemotherapy with Vincristine [23] or Lomustine (CCNU), Cisplatin, and Vincristine [24] in recurrent medulloblastoma, a large international prospective randomized trial (SIOP I) assessed the value of adjuvant chemotherapy with CCNU and Vincristine in the primary situation. All patients were treated by craniospinal irradiation. Adjuvant chemotherapy comprised Vincristine during radiotherapy and later CCNU and Vincristine in 6-weekly cycles for one year. The overall survival was 53% at five years and 45% at 10 years. At the end of the trial, the difference between the disease-free survival rate for the chemotherapy and control groups was statistically significant (*p* = 0.005). Later, relapses occurred in the chemotherapy arm and the difference between the two groups was lost. However, in the groups with partial or sub-total resection (*p* = 0.007), brainstem involvement (*p* = 0.001) and stages T3 and T4 disease (*p* = 0.002) benefit from chemotherapy persisted [11].

In parallel, the randomized Clinical Cancer Study Group Trial (CCSG-942) examined the efficacy of adjuvant chemotherapy following standard surgical treatment and radiation therapy in 233 patients. CSI was performed with or without adjuvant chemotherapy consisting of CCNU, Vincristine, and Prednisone. The estimated 5-year event-free survival (EFS) probability was not significantly different (59% for patients treated with CSI and chemotherapy and 50% for patients treated with radiation therapy only). In patients with more advanced tumors, event-free survival was longer after radiochemotherapy (48% vs. 0%, *p* = 0.006) and the survival time was significantly prolonged [12].

After these moderate initial experiences with Vincristine and CCNU, results of the more intensive combination of CCNU, Cisplatin, and Vincristine in combination with CSI appeared promising. Packer et al. reported results of large monocentric [25] and later multicentric [13] series comparing radiochemotherapy with radiotherapy alone. In both series, the 5-year survival rate for high-risk children was increased to >80% after CSI in combination with CCNU/VCR and Cisplatin. Later, in a multicentric prospective single arm trial, 65 children (3–10 years of age) with nondisseminated medulloblastoma were treated with postoperative, reduced-dose craniospinal radiation therapy (23.4 Gy) and 55.8 Gy of posterior fossa boost and the same chemotherapeutic regimen. Five-year PFS was 79%. [15]. In 2004, these results were confirmed by a larger prospective trial in which 421 patients (3–21 years of age) with nondisseminated medulloblastoma were treated with 23.4 Gy of CSI, 55.8 Gy of posterior fossa RT and Lomustine (CCNU), Cisplatin and Vincristine; or Cyclophosphamide, Cisplatin, and Vincristine. Five-year EFS and survival for the cohort of 379 assessable patients was 81% and 86%, respectively [2].

Later, more intensive approaches with hyperfractionated radiotherapy did not lead to a superior survival compared to conventional radiotherapy in average-risk medulloblastoma. In the large randomized SIOP PNET 4 trial, 340 children (4–21 years of age) received hyperfractionated radiotherapy (HFRT) or conventional fractionated radiotherapy, followed by eight cycles of Cisplatin, Lomustine, and Vincristine. Survival rates were not different between the two treatment arms with a 5-year overall survival (OS) of 87% and 85%, respectively [19]. Remarkably, patients with start of RT more than seven weeks after tumor resection had a worse prognosis [19].

Altogether, these studies set the current standard for average-risk medulloblastoma in childhood involving 23.4 Gy CSI with single doses of 1.8 Gy combined with CCNU, Cisplatin, and Vincristine.

### 3.3. High-Risk Medulloblastoma

Within the past decades, reduction in CSI doses has not been successfully implemented in high-risk patients. In trials, often prognostically different patients with relevant residual tumor or M1–M3 have been included (medulloblastoma staging system: Table 3), leading to very heterogenous survival data. With conventionally fractionated CSI radiotherapy to 36 Gy–39.6 Gy, several different chemotherapies have been tried. In the small CCG99701 trial, Carboplatin concomitantly to CSI and adjuvant chemotherapy led to a 5y-PFS of 70% [26]. In combination with intensive chemotherapy before radiotherapy in the phase III PNET3 trial, five year EFS (5y-EFS) was approximately 70% in patients with M0–M1 disease [17] and 35% in patients with M2–3 [3]. In the HIT 91 trial that combined CSI with Lomustine, Cisplatin, and Vincristine, long-term survival was also encountered in metastasized patients. The 10-year OS in M2 and M3 patients was 42% and 45%, respectively [18].

Hyperfractionated radiotherapy together with intensive pre- and postradiotherapy chemotherapy has also been implemented in a large prospective HIT2000 trial with a dose of 1.0 Gy twice daily (BID) to 40 Gy and 1.0 Gy BID boost to 28.0 Gy and a smaller series from Milano that used 1.3 Gy BID to 39.0 Gy and a boost to 60 Gy/1.5 Gy BID, leading to a five year EFS of 60% and 70%, respectively [27,28]. Due to the heterogeneity of data, to date, a clear treatment standard has not been set. In the future, molecularly based assignment to clinical trials will be performed to better clarify treatment effects in distinct genetic subgroups.

### 3.4. Adult Medulloblastoma

Due to the rarity of disease in adults and the absence of randomized trials, clinical evidence is less elaborate in adult medulloblastoma. In the prospective phase II NOA-07 trial, toxicity of conventionally fractionated CSI with an end dose 36.0 Gy has been evaluated in standard-risk patients together with concomitant treatment with Vincristine and adjuvant Lomustine and Cisplatin [29]. Here, all patients tolerated CSI well and 70% of patients tolerated at least four cycles of chemotherapy. Toxicity and feasibility appeared to be age-dependent, leading to the application of four cycles of chemotherapy in 73% of patients below age of 45 years and 63% of patients aged 45 or above (*p* = 0.66). Leukopenia, polyneuropathy, and ototoxicity were the most relevant toxicities [29].

In another phase II trial, Brandes et al. examined 26 adult high-risk patients in upfront chemotherapy with a MOPP-like regimen or with Cisplatin, Etoposide, and Cyclophosphamide. This treatment was followed by CSI and maintenance chemotherapy. PFS and OS rates at five years were 72% and 75%, respectively [30]. After indication of superiority in several cohorts of standard-risk and high-risk medulloblastoma [31,32,33,34], meta-analyses also pointed toward a significant survival benefit of radiochemo- compared to radiotherapy. Kocakaya et al. analyzed data from more than 907 patients. Patients who received radiochemotherapy had a significantly longer OS (median OS (mOS): 108 months) than patients treated with radiotherapy alone (mOS: 57 months) [35]. In an analysis of the National Cancer Database in UK, Kann et al. reported that radiochemotherapy was associated with a superior mOS compared with radiotherapy alone (HR: 0.53; 95% CI: 0.32–0.88, *p* = 0.01) [36].

Within most of the clinical trials from adult patients, conventionally fractionated CSI (single dose (SD) 1.8 Gy, end dose (ED) 36.0 Gy) with posterior fossa/tumor bed boost to 54.0 Gy) has been applied. Attempts to prospectively compare CSI dose reductions to 23.4 Gy are on the way in the international molecularly stratified EORTC 1634-BTG PersoMed I trial [37].

### 3.5. Timing of Radiotherapy

In two large prospective randomized trials, children who received pre-radiation chemotherapy had a significantly poorer EFS than those treated with immediate postoperative radiotherapy [33,34]. In the study of Bailey et al., 364 children were randomly assigned to receive or not receive pre-radiotherapy chemotherapy. Neither in low- nor high-risk patients, a benefit of pre-radiation chemotherapy was seen. Particularly in children receiving reduced dose radiotherapy and pre-irradiation chemotherapy, a poor outcome was observed [14]. In the German HIT91 trial, 137 patients were randomized between neoadjuvant and adjuvant chemotherapy. Neoadjuvant chemotherapy was accompanied by increased myelotoxicity of the subsequent radiotherapy, causing a higher rate of interruptions and an extended overall treatment time with a potential negative impact on outcome [16].

### 3.6. Neurotoxicity, Other Toxicity—And How to Avoid It

The clinical goal of medulloblastoma treatment is cure of disease, which is fortunately reached in many young patients. However, with cure and long-time survival, radiotherapy-related long-term toxicity is of major concern. Significant decline in neurocognitive and neuropsychological functioning after cranio- or craniospinal radiotherapy has been shown in many patient cohorts [38,39,40] and depends on the volume and dose of radiotherapy [40,41,42].

Apart from neurocognitive deficits, the pattern of long-term toxicity involves a variety of symptoms and issues. The Childhood Cancer Survivor Study (CCSS) compared 380 five-year survivors of medulloblastoma and 4031 siblings regarding the cumulative incidence of neurologic health conditions. Survivors were at increased risk of late-onset hearing loss (HR: 36.0), stroke (HR: 33.9), seizure (HR: 12.8), poor balance (HR: 10.4), tinnitus (HR: 4.8), and cataracts (HR: 31.8). Toxicity and long lasting symptoms were also reflected in biographic milestones: survivors were less likely than siblings to earn a college degree (relative risk [RR]: 0.49), marry (RR: 0.35) and live independently (RR: 0.58) [43].

In addition, radiotherapy of the spinal axis may decrease fertility and contribute to a risk of gonadal dysfunction and subsequent fertility issues in female patients caused by scatter irradiation [44].

#### 3.6.1. Boost Volume Reduction

As previously outlined, the major tool to decrease late detrimental effects is the reduction in the CSI radiation dose. Additionally, reduction in the boost volume from the posterior fossa to tumor bed is desirable in order to spare structures such as the inner ear and the temporal lobes/hippocampus. The large prospective ACNS0331 trial examined tumor bed vs. posterior fossa boost as a primary question and randomly assigned patients aged 3–21 years with average-risk medulloblastoma (MB) to receive posterior fossa radiation therapy (PFRT) or involved field radiation therapy (IFRT) following CSI. A total of 464 patients were eligible and evaluable to compare PFRT versus IFRT. The five-year EFS was 82.5% for IFRT and 80.5% for PFRT. IFRT was not inferior to PFRT (HR: 0.97; 94% upper CI, 1.32). The pattern of failure was not different between the two treatment arms. With current follow-up, smaller radiotherapy boost volumes did not affect long-term IQ. Children aged eight years and older at diagnosis treated with 23.4 Gy CSI exhibited no declines in IQ in the observed period following treatment [4].

#### 3.6.2. Advanced Techniques

In the past, the most commonly used treatment techniques in medulloblastoma were combinations of lateral opposing photon fields for the brain and posterior photon or electron beams for the spine, followed by a boost to the posterior fossa using lateral photon beams. During the last years, new 3D conformal treatment techniques for craniospinal irradiation were established. High precision photon techniques (e.g., volumetric arc therapy (VMAT) or tomotherapy) can reduce the dose to organs at risk outside the target volume and highly improve the speed and quality of treatment [45]. However, the volume of low dose irradiation in the body is increased [45,46,47,48], which needs to be taken into account for long-term issues such as secondary malignancies in other organs.

#### 3.6.3. Proton Treatment

Due to different physical properties, proton beam therapy can better spare normal tissue outside the target volume. Several comparative planning studies of conventional 3D conformal CSI and proton treatment show that radiation doses in normal tissues can be significantly reduced [49,50,51,52,53]. With this basis, radiobiological risk assessments from planning studies imply that, especially secondary cancer risk outside can be relevantly lowered with proton compared to photon/VMAT [54,55,56,57,58,59,60]. Larger comparative clinical data concerning this field are still sparse. In a retrospective series, 115 children with medulloblastoma received photon CSI (*n* = 63, group 1) or proton CSI (*n* = 52, group 2) followed by a boost to the tumor bed. The 5-year and 10-year secondary tumor incidence rates were 0.0% and 8.0%, respectively, in group 1; and 2.2% and 4.9%, respectively, in group 2; *p* = 0.74) [61]. However, data of larger controlled prospective clinical trials with long-term follow-up is needed to validate the findings on this important issue.

Furthermore, and with increasing clinical evidence, the risk of late toxic effects (e.g., endocrine dysfunction [62], cardiotoxicity [63,64], ototoxicity [65] and even neurotoxicity [66]) can potentially be reduced by proton use while maintaining similar tumor control compared to photon techniques [20,21] (Table 2).

Taking into account many advantages of the technique, proton beam therapy is increasingly being used, especially in children [67,68].

In further future innovations such as FLASH radiotherapy, which is currently still in the very early phase of development, could widen the therapeutic effect of radiotherapy in medulloblastoma [69,70,71].

### 3.7. The Influence of Quality Assurance

As outlined during the past decades, post-surgery therapy of medulloblastoma changed from high-dose craniospinal irradiation for all patients to risk-adapted treatment schedules including combinations of radiotherapy and chemotherapy, different craniospinal dose levels, and/or radiotherapy fractionation schemes. The main aim of all these modifications was to reduce late toxicity and improve post-treatment quality of life without compromising tumor control. Correct interpretation of all staging modalities (MRI imaging, cerebrospinal fluid examination, histological and molecular features) is thereby indispensable to ensure correct treatment stratification. Inappropriate treatment schedules can lead to worse tumor control as retrospectively shown in the Children’s Oncology Group (COG) A9961 study [2]. Therefore, quality control of staging and treatment stratification is important in medulloblastoma. Central review systems such as those used in prospective trials or as described by the German HIT-network appear to be beneficial [19,28,72].

Inadequate radiotherapy treatment fields have been common in retrospective central plan review studies and can potentially lead to worse tumor control in medulloblastoma, as shown in cohorts treated by simulation based radiotherapy [17,73,74,75]. Therefore, pre-treatment quality control of radiotherapy plans was introduced in clinical trials [76]. However, RT techniques changed from simulation-based treatment fields based on reference bony structures to 3D conformal radiotherapy in particular high precision photon techniques (e.g., intensity modulated radiotherapy, tomotherapy) or proton beam therapy. First experiences of pre-treatment quality assurance in patients treated by 3D conformal radiotherapy within the SIOP-PNET5 MB trial in Italy, Germany, Austria, and Switzerland showed high rates of plan deviations. Plan modifications were recommended in approximately 40% of evaluated plans [72,77,78,79]. The SIOPE published a consensus guideline on craniospinal target volume delineation to improve the quality of contouring in high-precision radiotherapy [80]. However, a pre-treatment quality assurance program is desirable to ensure correct application of radiotherapy in or even outside multicenter trials, especially in the case of decentralized treatment.

### 3.8. Risk Adapted Radiochemotherapy—Future Perspectives

In 2010, genetically defined subgroups of medulloblastoma were introduced dividing medulloblastoma into four different entities (wnt-activated, SHH-activated with p53 wild-type/SH-activated with mutated p53 Group 3 and Group 4) with distinct characteristics and prognoses [81]. The WHO classification of CNS-tumors adapted this subgrouping approach [82]. Consequently, the historic risk stratification system relying on the Chang staging system [83,84] (Table 3) with the risk factors residual disease >1.5cm^2^, metastatic dissemination, and large-cell/anaplastic histology needed to be reconsidered. In 2016, a consensus paper was published suggesting risk groups integrating medulloblastoma subgroups, specific prognostic genetic alterations, and the metastatic state of disease [85] (Table 4).

The importance of molecularly based subgroups has later been underlined in several analysis (e.g., in a post hoc analysis of a large cohort of children and adults with metastatic disease) and available tissue material 5-year EFS and OS differed between low-risk (WNT, *n* = 4; both 100%), high-risk (MYCC/MYCN amplification; *n* = 5, both 20%) and intermediate-risk patients (neither; *n* = 72, 63% and 73%), respectively [27].

Being a constant work in progress, new disease subgroups and risks groups have been implemented in the analysis of ongoing clinical trials concerning therapy reduction in low-risk and treatment intensification in high-risk patients. Concerning the latter, a prospective randomized trial for high-risk patients from the Children’s Oncology Group involving concomitant Carboplatin during radiotherapy showed a survival benefit in the Carboplatin arm in group 3 patients only [86].

With regard to de-intensification of treatment, a CSI dose reduction to 18.0 Gy was applied in the ACNS0331 trial in a randomized controlled fashion in average-risk patients. Unfortunately, this approach led to inferior EFS rates in the dose-reduced CSI arm, mainly driven by a significant difference in the group 4 patients, while the other groups did not show a significant EFS difference between CSI dose of 18.0 vs. 23.4 Gy [4].

Currently, recruiting trials such as the PNET5 trial (NCT02066220) and the SJMB12 (NCT01878617) trial apply integrated genetically pre-defined risk groups in order to prospectively tailor treatment. The aim of the low-risk treatment arm in PNET5 is to confirm the high rate of event-free survival in non-infants with standard-risk medulloblastoma with a low-risk biological profile (total or near-total tumor resection, non-metastatic and low-risk biological profile, defined as ß-catenin nuclear immuno-positivity) after CSI to 18.0 Gy and a tumor bed boost to 54 Gy.

In the multi arm phase II SJMB12 trial, patients will be stratified based on both clinical risk (low-, standard-, intermediate-, or high-risk) and molecular subtype (WNT, SHH, or Non-WNT Non-SHH). The stratified clinical and molecular treatment approach will be used to evaluate whether radiochemotherapy reduction is possible in low-risk WNT tumors and if targeted chemotherapy with sonic hedgehog inhibition after standard chemotherapy will induce benefits in SHH positive tumors. Additional new chemotherapy agents are added to standard chemotherapy to improve the outcome for intermediate- and high-risk Non-WNT/Non-SHH tumors.

### 3.9. New Combinatorial Approaches in High-Risk Medulloblastoma

Several strategies that appear promising from pre-clinical results could be applied to tackle resistance to radiochemotherapy in high-risk patients. First candidate drugs comprise inhibitors of DNA damage repair/cell cycle checkpoint inhibitors [87,88]. For instance, Prexacertib (LY2606368), a novel Chk inhibitor with preference for Chk1, is being tested at present in clinical trials for medulloblastoma together with Gemcitabine as a DNA damage inducer (NCT04023669). Within this treatment concept, inactivation of G2M checkpoint kinase 1 (Chk1) by Prexacertib blocks the homologous recombination, a major DNA damage repair pathway, and facilitates mitotic cell death. Particularly in p53-functional deficient tumors that are lacking the p53-dependend alternative repair pathway by non-homologous end-joining, this strategy could prove to be efficient and highly tumor specific.

Besides DNA damage induction, radiation simultaneously activates pro-survival pathways, often mediated by receptor tyrosine kinases that may prevent cells from undergoing apoptosis. Targeting these pathways has the potential to overcome radioresistance (reviewed in [89]).

Alternative approaches may involve compounds tackling tumor promoting pathways deregulated in medulloblastoma. Epigenetic regulators could release silenced tumor suppressor genes that are aberrantly hypermethylated in 70–90% of primary medulloblastoma [90]. In addition, telomerase inhibitors might sensitize MB cells to radio-chemotherapy [91,92] as enhanced telomerase activity is seen in over 70% of medulloblastoma [93].

Finally, the role of combining radiochemotherapy with immune therapy, (tumor specific vaccination, immune-checkpoint inhibitors such as peripheral death (PD1/PDL1), oncolytic viral therapy) remains to be defined in the future (reviewed in [94,95]). Currently, a clinical trial with mRNA-based vaccine is ongoing in adults with recurrent MB and primitive neuroectodermal tumors (NCT01326104).

In order to tailor therapy to the patients, tumor gene expression and high-throughput drug response data using orthotopic patient-derived xenografts (PDX), tailoring therapy based on the molecular and cellular characteristics of patients’ tumors [96] may be a valuable new option.

## 4. Conclusions

Radiotherapy is a powerful treatment tool in medulloblastoma with regard to efficacy, but can also induce significant toxicity. Up to now, treatment has evolved providing a curative approach for low- to standard-risk patients by using technically advanced dose-reduced CSI with a tumor bed boost in combination with polychemotherapy. In the future, subgroup-based adaptations of treatment in innovative combinations to further increase efficacy and to spare toxicity are needed.

## Figures and Tables

**Table 3 cancers-13-05945-t003:** Modified medulloblastoma staging system according to Chang.

T-Stage	Tumor Extent
T1	Tumor less than 3 cm in diameter
T2	Tumor greater than 3 cm in diameter
T3a	Tumor greater than 3 cm in diameter with extension into the aqueduct of Sylvius and/or the foramen of Luschka
T3b	Tumor greater than 3 cm in diameter with unequivocal extension into the brain stem
T4	Tumor greater than 3 cm in diameter with extension up past the aqueduct of Sylvius and/or down past the foramen magnum
**M-Stage**	**Degree of Metastasis**
M0	No evidence of gross subarachnoid or hematogenous metastasis
M1	Microscopic tumor cells found in the cerebrospinal fluid
M2	Gross nodular seeding demonstrated in the cerebellar/cerebral subarachnoid space or in the third or lateral ventricles
M3	Gross nodular seeding in the spinal subarachnoid space
M4	Metastasis outside the cerebrospinal axis

**Table 4 cancers-13-05945-t004:** Molecular based risk groups in medulloblastoma.

Risk	5y OS	Characterization
Low	>90%	WNT subgroup and non-metastatic group 4 tumors with whole chromosome 11 loss or whole chromosome 17 gain
Average	75–90%	Non-metastatic SHH TP53wt without MYCN amplification, non-metastatic group 3 without MYCN amplification, non-metastatic group 4 with intact chromosome 11
High	50–75%	Metastatic SHH or group 4 tumors, or MYCN amplified SHH medulloblastoma
Very High	<50%	Group 3 with metastases or SHH with TP53 mutation

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
