# Peer review of "Radiotherapy in Medulloblastoma—Evolution of Treatment, Current Concepts and Future Perspectives"

_cancers, 2021, doi:10.3390/cancers13235945_

Round 1
Reviewer 1 Report
This is a nicely prepared review - that highlights progress in the treatment of Medulloblastoma. It was informative and to the point. I have but a few minor suggestions.
The section on proton therapy was disappointingly sparse - and could easily include an additional table that highlight the reduction in normal tissue toxicities realized by overlapping proton treatment plans. In this light - this could also be compared to current VMAT treatment plans to highlight the pros and cons of these 2 state-of the art MB radiotherapy treatment options.
In terms of the advanced techniques section - I would be curious to see a section (if not speculation) on whether the authors envision a potential role in FLASH radiotherapy for the treatment of MB. From what I read about this technique, it may be an better option for addressing the high-risk stratified patients.
Author Response
informative and to the point. I have but a few minor suggestions.
The section on proton therapy was disappointingly sparse - and could easily include an additional table that highlight the reduction in normal tissue toxicities realized by overlapping proton treatment plans. In this light - this could also be compared to current VMAT treatment plans to highlight the pros and cons of these 2 state-of the art MB radiotherapy treatment options.
Thank you for this valuable comment. We agree that the role of proton treatment should be discussed in more detail and rewrote the proton paragraph:
“Due to different physical properties proton beam therapy can better spare normal tissue outside the target volume. Several comparative planning studies of conventional 3D conformal CSI and proton treatment show that radiation doses in normal tissues can be significantly reduced [46–50]. With this basis, radiobiological risk assessments from planning studies imply that especially secondary cancer risk outside can be relevantly lowered with proton compared to photon/VMAT [51–57]. Larger comparative clinical data concerning this field is still sparse. In a retrospective series 115 children with medullo-blastoma received photon CSI (n = 63, group 1) or proton CSI (n= 52, group 2) followed by a boost to the tumor bed. The 5-year and 10-year secondary tumor incidence rates were 0.0% and 8.0%, respectively, in group 1; and 2.2% and 4.9%, respectively, in group 2; p=0.74) PMID: 34254296. However, data of larger controlled prospective clinical trials with long-term follow up is needed to validate findings about this important issue.
Further, and with increasing clinical evidence, the risk of late toxic effects, e.g. endo-crine dysfunction [58], cardiotoxicity [59,60], ototoxicity [61] and even neurotoxicity [62] can potentially be reduced by proton use while maintaining similar tumor control com-pared to photon techniques [63,64], Table 2.
Taking into account many advantages of the technique, proton beam therapy is increasingly used especially in children [65–66].”
We were hesitant to include an additional table as we mainly reserved the tables for results of randomized trials.
In terms of the advanced techniques section - I would be curious to see a section (if not speculation) on whether the authors envision a potential role in FLASH radiotherapy for the treatment of MB. From what I read about this technique, it may be an better option for addressing the high-risk stratified patients.
Thank you for this comment, we added a small section after proton treatment regarding this innovative technique – more thought provoking, not going into detail on this:
“In the further future innovations like FLASH radiotherapy, which is currently still in the very early phase of development could widen the therapeutic effect of radiotherapy in medulloblastoma (PMID: 31829765, PMID: 32899466, PMID: 31253466).”
Reviewer 2 Report
Paper is interesting, well argued and fluid in writing.
There are some typos to correct, please review it carefully. In the paragraph of adult medulloblastoma it would be useful to better specify the stage of disease of adults in the reported studies.
The citations of bibliography do not correspond: they are cited up to number 83 but in the bibliography they end at 65. Please review all bibliography.
Citation 56 speaks about the pattern of failure after photons in medulloblastoma but in the paragraph "proton treatment" only the aspect of late toxicity improvement is argued; it would be useful to argue risks and benefit in a deeper way.
Author Response
Paper is interesting, well argued and fluid in writing.
There are some typos to correct, please review it carefully. In the paragraph of adult medulloblastoma it would be useful to better specify the stage of disease of adults in the reported studies.
Thank you for your justified comments, we corrected typos, stage of disease (standard or high risk) was added to the reported studies.
The citations of bibliography do not correspond: they are cited up to number 83 but in the bibliography they end at 65. Please review all bibliography.
Thank you very much, the bibliography was incompletely refreshed, we changed this and reviewed the bibliography.
Citation 56 speaks about the pattern of failure after photons in medulloblastoma but in the paragraph "proton treatment" only the aspect of late toxicity improvement is argued; it would be useful to argue risks and benefit in a deeper way.
Thank you, we agree that the paragraph regarding proton treatment was not very balanced. We changed the structure and rewrote the paragraph (please see above).
We thank all reviewers for their valuable advice to increase the quality of the manuscript.